# Antiproliferative and Cytotoxic Cytochalasins from *Sparticola triseptata* Inhibit Actin Polymerization and Aggregation

**DOI:** 10.3390/jof8060560

**Published:** 2022-05-25

**Authors:** Katherine Yasmin M. Garcia, Mark Tristan J. Quimque, Christopher Lambert, Katharina Schmidt, Gian Primahana, Theresia E. B. Stradal, Andreas Ratzenböck, Hans-Martin Dahse, Chayanard Phukhamsakda, Marc Stadler, Frank Surup, Allan Patrick G. Macabeo

**Affiliations:** 1The Graduate School, University of Santo Tomas, España Blvd., Manila 1015, Philippines; katherineyasmin.garcia.gs@ust.edu.ph (K.Y.M.G.); mtjquimque@gmail.com (M.T.J.Q.); 2Laboratory for Organic Reactivity, Discovery and Synthesis (LORDS), Research Center for the Natural and Applied Sciences, University of Santo Tomas, España Blvd., Manila 1015, Philippines; 3Chemistry Department, College of Science and Mathematics, Mindanao State University–Iligan Institute of Technology, Tibanga, Iligan City 9200, Philippines; 4Department of Microbial Drugs, Helmholtz Centre for Infection Research and German Centre for Infection Research (DZIF), Partner Site Hannover/Braunschweig, Inhoffenstraße 7, 38124 Braunschweig, Germany; christopher.lambert@helmholtz-hzi.de (C.L.); gian.primahana@lipi.go.id (G.P.); marc.stadler@helmholtz-hzi.de (M.S.); 5Department of Cell Biology, Helmholtz Centre for Infection Research (HZI), Inhoffenstraße 7, 38124 Braunschweig, Germany; katharina.schmidt@helmholtz-hzi.de (K.S.); theresia.stradal@helmholtz-hzi.de (T.E.B.S.); 6Research Center for Chemistry, National Research and Innovation Agency (BRIN), Kawasan Puspitek, Serpong, Tangerang Selatan 15314, Indonesia; 7Institut für Organische Chemie, Universität Regensburg, Universitätstrasse 31, 93053 Regensburg, Germany; andreas.ratzenboeck@chemie.uni-regensburg.de; 8Leibniz-Institute for Natural Product Research and Infection Biology, Hans Knöll Institute (HKI), 07745 Jena, Germany; hans-martin.dahse@hki-jena.de; 9Center of Excellence in Fungal Research, Mae Fah Luang University, Chiang Rai 57100, Thailand; chayanard91@gmail.com; 10Institute of Plant Protection, College of Agriculture, Engineering Research Center of Chinese Ministry of Education for Edible and Medicinal Fungi, Jilin Agricultural University, Changchun 130118, China; 11Institute of Microbiology, Technische Universität Braunschweig, Spielmannstraße 7, 38106 Braunschweig, Germany

**Keywords:** *Sparticola triseptata*, structure elucidation, ECD–TDDFT, antiproliferative, cytotoxic, actin inhibitors

## Abstract

Laying the groundwork on preliminary structure–activity relationship study relating to the disruptive activity of cytochalasan derivatives on mammalian cell actin cytoskeleton, we furthered our study on the cytochalasans of the Dothideomycetes fungus, *Sparticola triseptata*. A new cytochalasan analog triseptatin (**1**), along with the previously described cytochalasans deoxaphomin B (**2**) and cytochalasin B (**3**), and polyketide derivatives *cis*-4-hydroxy-6-deoxyscytalone (**4**) and 6-hydroxymellein (**5**) were isolated from the rice culture of *S. triseptata*. The structure of **1** was elucidated through NMR spectroscopic analysis and high-resolution mass spectrometry (HR-ESI-MS). The relative and absolute configurations were established through analysis of NOESY spectroscopic data and later correlated with experimental electronic circular dichroism and time-dependent density functional theory (ECD–TDDFT) computational analysis. Compounds **1** and **2** showed cytotoxic activities against seven mammalian cell lines (L929, KB3.1, MCF-7, A549, PC-3, SKOV-3, and A431) and antiproliferative effects against the myeloid leukemia K-562 cancer cell line. Both **1** and **2** were shown to possess properties inhibiting the F-actin network, prompting further hypotheses that should to be tested in the future to enable a well-resolved concept of the structural implications determining the bioactivity of the cytochalasin backbone against F-actin.

## 1. Introduction

Cytochalasans are a diverse group of biologically active fungal polyketide–amino acid hybrids featuring a tricyclic core structure composed of a 7- to 15-membered macrocycle fused to a highly substituted perhydroisoindolone moiety. They are biosynthesized via a PKS-NRPS hybrid pathway [1]. Since the first report of cytochalasins A and B from *Phoma* strain S 298 and *Helminthosporium dematioideum*, over 300 derivatives have been reported in several genera of Dothideomycetes, such as *Ascochyta*, *Preussia*, and *Phoma* [2,3,4], although producers mainly belong to Sordariomycetes, namely *Aspergillus*, *Daldinia*, and *Diaporthe* (formerly known as *Phomopsis* and *Xylaria*) [5,6,7]. Variations in the amino acid side chain, intramolecular rearrangements, and observation of unique substitution patterns in the macrocyclic ring have led to a diversity of cytochalasin structures. Cytochalasans exhibit antimicrobial, antiviral, and antiparasitic properties [8,9,10,11,12], regulate hormonal functions [13,14], inhibit cholesterol synthesis [15] and bacterial biofilms [16], and interfere with glucose transport proteins [17] and Ca^2+^ influx regulation [18]. By far, the most prominent and frequently reported biological effects of cytochalasans have been related to their interference with the actin cytoskeleton [5]. Interestingly, different derivatives of this compound class can either have strong and irreversible or weak and reversible activities. Studies on the structure–activity relationship of such mechanism are limited [6].

We have previously reported the actin depolymerization activity of 25 cytochalasans isolated from ascomata and mycelial cultures of different Ascomycota to establish a preliminary structure–activity relationship study [5]. We noted that the presence of hydroxyl group in the C7 and C18 of the cytochalasan backbone and their stereochemical configurations are important factors for actin cytoskeleton polymerization inhibition. When the reversibility of the actin-disrupting effects was evaluated, no direct correlations between potency and reversibility in the tested compound group were observed. As part of our ongoing efforts to explore biologically active Dothideomycetes secondary metabolites, the ascomycete *Sparticola triseptata* (Leuchtm.) Phukhamsakda & K. D. Hyde, *comb. nov*. obtained from the decomposing branches of *Tofielda calyculata* (L.) Wahlenb. was investigated for its cytotoxic chemical constituents. In this study, we carried out the isolation and structure elucidation of one cytotoxic cytochalasin derivative from the solid rice culture of *S. triseptata*, hitherto referred to as triseptatin (**1**), along with deoxaphomin B (**2**), cytochalasin B (**3**), *cis*-4-hydroxy-6-deoxyscytalone (**4**), and 6-hydroxymellein (**5**). To complement and further investigate the cytotoxic effect on the F-actin network of mammalian cell lines of cytochalasin **1**–**3**, 1 h endpoint assays using fluorescence microscopy were also carried out.

## 2. Materials and Methods

### 2.1. General Experimental Procedures

Specific optical rotations ([α]^D^) were measured on a Perkin Elmer 241 polarimeter in a 100 mm × 2 mm cell at 20 °C. Nuclear magnetic resonance (NMR) spectra were obtained either on a Bruker Ascend 600 MHz spectrometer equipped with a 5 mm TXI cryoprobe (^1^H 600 MHz, ^13^C 150 MHz) or a Varian VNMRS-500 MHz (^1^H 500 MHz, ^13^C 125 MHz). Spectra were acquired at 25 °C (unless otherwise specified) in MeOH-*d*_4_ with reference to residual ^1^H or ^13^C signals in the deuterated solvent. HR-ESI mass spectra were measured using Agilent 6200 series TOF and 6500 series Q-TOF LC/MS systems. The HPLC-DAD purification was performed on a Shimadzu Prominence liquid chromatograph LC-20AT coupled with a SPD-M20A photodiode array detector (Shimadzu Corp., Tokyo, Japan) and the semipreparative reversed-phase C18 column Inertsil ODS-3 (10 mm I.D. × 250 mm, 5 μm, G.L. Sciences, Tokyo, Japan). The mobile phase was composed of ultrapure water (Milli-Q, Millipore, Schwalbach, Germany) as solvent A and acetonitrile (HPLC grade) as solvent B.

### 2.2. Fungal Material

The ascomycete *Sparticola triseptata* (Leuchtm.) Phukhamsakda & K.D. Hyde, which represents the ex-type strain of the species, was isolated from a decayed branch of *Tofieldia calyculata* (L.) Wahlenb. [19]. Phukhamsakda et.al [20] recently transferred this species to the genus *Sparticola* based on a polyphasic taxonomic study that included molecular phylogenetic and morphological methods. The type strain is deposited at the KNAW Westerdijk Fungal Biodiversity Centre, Utrecht, the Netherlands (CBS 614.86).

### 2.3. Production, Extraction, Isolation, and Structural Characterization

The fungal strain was cultured on malt extract agar plates for 8–10 weeks until the culture developed its characteristic brownish pigmentation. The fungus was cultivated on a solid rice media (70 g brown rice, 0.3 g peptone, 0.1 g corn syrup, and 100 mL ultrapure water) in 15 × 1000 mL sterilized Fernbach culture flasks, followed by autoclaving (121 °C, 20 min). Five agar blocks of a well-grown fungal culture were inoculated in the culture flasks and incubated under static condition in a dark room at 25–30 °C for 12 weeks until the fungal hyphae proliferated and the rice medium turned black in color. The rice cultures were homogenized using a sterile metal spatula. Fermentation was terminated by the addition of ethyl acetate (EtOAc; 3 × 300 mL). The combined extracts were concentrated in a rotary evaporator to afford the crude extract (30 g). The crude EtOAc extract was reconstituted with 300 mL 10% aqueous methanol and partitioned with *n*-heptane (3 × 100 mL). The combined organic layer was concentrated *in vacuo* to afford a dark brown methanolic crude extract (8.8 g).

The methanolic crude extract was fractionated using silica gel column chromatography, and elution was carried out using the following solvent systems: petroleum ether-EtOAc (1:1, 2:3, 3:7, 1:4, 1:9), EtOAc, dichloromethane (DCM), DCM–MeOH (9:1, 1:4, 7:3, 3:2, 1:1), and methanol to afford five combined main fractions. Fraction 3 (3.40 g) was chromatographed with DCM–MeOH (5:1) to yield three subfractions. The second subfraction (3.29 g) was subsequently eluted with DCM–MeOH (40:1) to afford five subfractions. The third subfraction, fraction 3.2.3 (259 mg), was further purified using semipreparative RP-HPLC. The mobile phase was composed of ultrapure water (Milli-Q, Millipore, Schwalbach, Germany; solvent A) and acetonitrile (RCI Labscan Ltd., HPLC grade; solvent B). Purification was carried out using the following gradient: 40% solvent B for 5 min, 40–100% solvent B for 20 min, 100% solvent B for 5 min, and 100–40% solvent B for 5 min. This resulted in 13 fractions affording **3** (2.12 mg, flow rate = 4.0 mL min^−1^, UV detection 200−600 nm, *t*_R_ = 10.03 min), **4** (2.30 mg, flow rate = 4.0 mL min^−1^, UV–vis detection 200−600 nm, *t*_R_ = 15.69 min), and **2** (6.87 mg, flow rate = 4.0 mL min^−1^, UV–vis detection 200−600 nm, *t*_R_ = 22.85 min). Fraction 3.2.1 (1.54 g) was further chromatographed using petroleum ether–EtOAc (2:1, 1:1) and DCM–MeOH (100:1, 80:1) to yield three homogenous subfractions. The last subfraction, fraction 3.2.1.3 (889 mg), was subjected to semipreparative reversed-phase HPLC using similar gradient solvent system composition as described above to yield 10 fractions. Fraction 3.2.1.3.7 (53.8 mg) was purified twice by employing a linear gradient of solvent B from 40% solvent B for 5 min, 40–100% solvent B for 25 min, 100% solvent B for 5 min, and 100–40% solvent B for 5 min to yield compound **2** (7.64 mg, flow rate = 4.0 mL min^−1^, UV detection 200−600 nm, *t*_R_ = 14.24 min). Fraction 3.2.1.3.10 (7.58 mg) was also subjected to semipreparative reversed-phase HPLC using similar gradient conditions to afford **1** (3.65 mg, 4.0 mL min^−1^, UV detection 200−600 nm, *t*_R_ = 18.65 min).

Triseptatin (**1**): white amorphous solid; [α]_D_^25^ +30 (*c* 0.1, MeOH); UV (*c* 0.1, MeOH) λ_max_ (log ε) 258 (3.83), 289 (3.89) nm; HPLC–ECD data in acetonitrile as λ_max_ (Δε) 205 (0.0074), 216 (–0.0025), and 239 (–0.0011). ^1^H and ^13^C NMR data, Table 1; HR-ESI-MS *m*/*z*: [M + H]^+^ calcd for C_31_H_40_NO_5_, 506.2906; found, 506.2901 (Appendix A). IUPAC nomenclature: (3*S*,5*S*,7*S*,8a*R*, 9a*R*,13*E*,16*R*,20*R*,21*E*)-3-benzyl-6-hydroxy-4,10-dimethyl-5-methylene-1,17-dioxo-2,3,3a,4,5,6,6a,9,10,11,12,13,14,17-tetradecahydro-1H-cyclotrideca[d]isoindol-14-yl acetate.

### 2.4. Computational Calculations

A conformational analysis on triseptatin (**1**) was performed using the Avogadro (version 1.1.1) platform [21], which included a search for low-energy conformations using the MMFF94 molecular mechanics force field and conformer optimization following the steepest descent algorithm. All stable conformations were subjected to geometry reoptimizations via density functional theory calculations with B3LYP/6-31G(d) basis set on a polarizable continuum model (PCM) with methanol as the solvent model. The calculated energies, taken as the sum of electronic and zero-point energies, were used to estimate the Boltzmann population for each conformer. The optimized geometries were then subjected to time-dependent DFT (TDDFT) following the same functional, basis set, and PCM solvent model. A Gaussian distribution function was used to generate the ECD curve from the calculated rotatory strength values with 3000 cm^−1^ half-height width. All DFT calculations were carried out using Gaussian 16W [22], while the visualization of results was conducted on GaussView 6.0.

### 2.5. Antiproliferation and Cytotoxicity Assays

Compounds **1**–**3** were assayed against human umbilical vein endothelial cells (HUVEC) and K-562 human chronic myeloid leukemia cells (DSM ACC 10) for their antiproliferative effects (GI_50_) [23]. Cytotoxicity properties of the compounds were also assessed against several mammalian cancer cell lines, including mouse fibroblast L929, HeLa (KB3.1), human breast adenocarcinoma (MCF-7), adenocarcinomic human alveolar basal epithelial cells (A549), human prostate cancer cells (PC-3), ovarian carcinoma (SKOV-3), and squamous cell carcinoma (A431), and expressed as IC_50_ in the MTT assay [24]. Inhibitory concentrations were calculated as 50% half-maximal inhibitory concentration (IC_50_, concentration of the substance where a specific biological process is reduced by half), 50% inhibition of cell growth (GI_50_, the concentration needed to reduce the growth of treated cells to half that of untreated cells), or 50% cytotoxic concentration (CC_50_, the concentration that kills 50% of treated cells).

### 2.6. Cell Culture of U2-OS Cells and Actin Disruption Assay

The impact of **1** and **2** on the organization of filamentous actin in tissue culture cells was investigated in an actin disruption assay following the procedure presented by [6]. Cytochalasin B (**3**, previously isolated by [6]) and D (MP Biomedicals, Solon, OH, USA) were chosen as widely used controls for cytochalasan impact on F-actin network organization in DMSO (Carl Roth GmbH, Karlsruhe, Germany). DMSO was also used as a vehicle control. Concentrations to study the effects on the F-actin organization were estimated based on previously determined IC_50_ against L929 mouse fibroblasts (1 × IC_50_ and 5 × IC_50_, low-dose and high dose (cf. Wang et al. [25]). Concurrently, a second testing strategy consisted of a titration experiment employing concentrations ranging from 30 to 0.03 µM of compounds **1**, **2**, and **3**. Cells of the osteosarcoma cell line U2-0S [ATCC HTB-96] were cultured in Dulbecco’s modified minimum essential medium (DMEM, Life Technologies, Carlsbad, CA, USA) containing 10% fetal bovine serum (Sigma-Aldrich, St. Louis, MO, USA), 1% L-glutamine (Life Technologies, Carlsbad, CA, USA), 1% minimum essential medium nonessential amino acids (MEM NEAA, Life Technologies, Carlsbad, CA, USA), and 1% sodium pyruvate (Life Technologies, Carlsbad, CA, USA) at 37 °C and 7.5% CO_2_ for the following experiments. Fibronectin (Roche, Mannheim, Germany)-coated cover slips were seeded with 20,000 cells and allowed to spread overnight and treated with overnight equilibrated culture medium spiked with cytochalasins or DMSO in the abovementioned concentrations. Cells were treated with the prepared medium and incubated for 1 h before being washed once with PBS (pH 7.4) and fixed with 4% prewarmed paraformaldehyde (AppliChem, Darmstadt, Germany) supplied in PBS for 20 min. Reversibility of the impact on the F-actin network organization was tested by washout with 3× prewarmed PBS prior to the addition of fresh medium and a recovery time of 1 h before fixation. Fixed cells were washed with PBS thrice and permeabilized using 0.1% Triton X-100 (Bio-Rad Laboratories, Hercules, CA, USA) in PBS for 1 min at room temperature. After three additional washing steps with PBS, fluorescently labelled phalloidin (ATTO-594, ATTO-Tec, Siegen, Germany) diluted in PBS (1:100) was used and incubated for an additional hour to probe for the F-actin cytoskeleton and mounted in ProLong Diamond Antifade Mountant (Invitrogen, Carlsbad, CA, USA) containing DAPI to probe for nuclear DNA. Samples were examined for epifluorescence using an inverted microscope (Nikon eclipse Ti2, Tokio, Japan) with a 60× Nikon oil immersion objective (Plan Apofluar, 1.4 NA) with pE-4000 (CoolLED, Andover, UK) as a light source. Pictures were recorded with a pco.edge back-illuminated sCMOS camera (Excelitas Technologies, Mississauga, ON, Canada) operated by NIS elements (Nikon, Tokio, Japan) and processed by Image J (NIH, Bethesda, MD, USA).

## 3. Results and Discussion

The EtOAc extract of *Sparticola triseptata* obtained from the rice culture was partitioned between *n*-heptane and 10% aqueous MeOH. The resulting aqueous methanolic crude extract was purified using silica gel column chromatography and semipreparative HPLC to afford metabolites **1**–**5** (Figure 1). The known compounds, deoxaphomin B (**2**), cytochalasin B (**3**), *cis*-4-hydroxy-6-deoxyscytalone (**4**), and 6-hydroxymellein (**5**), were identified by comparing their physicochemical and NMR spectroscopic data with those reported in the literature [4,26,27,28,29].

Triseptatin (**1**) was obtained as an optically active whitish-yellow amorphous solid. The molecular formula C_31_H_40_NO_5_, indicating 13 degrees of unsaturation, was established based on the protonated molecular ion peak at *m/z* C_31_H_39_NO_5_ [M + H]^+^ of its positive-ion HR-ESI-MS. This was consistent with the number of proton and carbon peaks detected in its ^1^H and ^13^C NMR spectroscopic data (Table 1). Preliminary spectroscopic analysis of its 1D NMR data showed identical resonances to that of deoxaphomin, which was first isolated from the organic culture extracts of *Ascochyta heteromorpha* and *Phoma multirostrata*.

The ^1^H NMR spectrum of **1** (Appendix A) displayed three methyl proton signals at *δ_H_* 0.79 (3H, *d*, *J* = 6.8 Hz, H-11), 0.89 (3H, *d*, *J* = 6.9 Hz, H-24), and 2.04 (3H, s, H-26); two exocyclic methylene proton signals at *δ_H_* 5.03 (1H, *s*, H-12a) and 5.17 (1H, *s*, H-12b); four olefinic proton signals at *δ_H_* 5.22 (1H, *m*, H-14), 6.01 (1H, *dd**d*, *J* = 15.2, 9.6, 1.9 Hz, H-13), 6.48 (1H, *dd*, *J* = 15.4, 8.5 Hz, H-21), and 6.86 (1H, *d*, *J* = 15.4 Hz, H-22); two pairs of *ortho*-coupled aromatic proton signals at *δ_H_* 7.08 (2H, *d*, *J* = 7.6 Hz, H-2′/H-6′) and 7.32 (2H, *t*, *J* = 7.6 Hz, H-3′/H-5′); and several aliphatic proton signals between *δ_H_* 1.09 and 3.38 (Table 1). The ^13^C and heteronuclear single quantum coherence–distortionless enhancement by polarization transfer (HSQC–DEPT) NMR spectroscopic data revealed the presence of three carbonyl carbons, one quaternary carbon, two sp^2^ nonprotonated carbons, five aromatic methines, four olefinic methines, seven sp^3^ methines (two oxygenated and five nonoxygenated), one sp^2^ methylene, five sp^3^ methylene carbons, and three methyl carbons.

Analysis of 1D NMR spectroscopic data revealed that **1** possessed a cytochalasin core skeleton, which was confirmed by its homonuclear and heteronuclear 2D NMR data (Appendix A). The ^1^H–^1^H correlation spectroscopy (COSY) cross peaks of *δ_H_* 3.38 (1H, *m*, H-3) with *δ_H_* 2.91 (1H, *m*, H-4), *δ_H_* 2.46 (1H, *dd*, *J* = 13.0, 7.3 Hz, H-10a), and *δ_H_* 2.64 (1H, *dd*, *J* = 13.0, 6.4 Hz, H-10b), as well as the long-range HMBC coupling from H-3 to *δ_C_* 176.4 (C-1) and *δ_C_* 46.3 (C-4), from H-4 to C-1 and *δ_C_* 63.9 (C-9), and from H_2_-10 to *δ_C_* 54.0 (C-3), revealed the presence of a 2-pyrrolidinone fragment. Key HMBC correlations from H-4 to *δ_C_* 33.2 (C-5) and *δ_C_* 151.0 (C-6); from H_3_-11 to C-4, C-5, and C-6; from the exocyclic methylene H_2_-12 to C-5 and *δ_C_* 72.8 (C-7); and from *δ_H_* 2.50 (1H, *t*, *J* = 9.9 Hz, H-8) to *δ_C_* 63.9 (C-9) established the linkage between the methylidenecyclohexanol ring and 2-pyrrolidinone moiety forming the perhydroisoindolone subunit (Figure 2). Further analyses of the COSY spectrum led to the construction of an *ortho*-coupled (*J* = 7.6 Hz) three distinct proton spin systems illustrating homonuclear coupling correlations between H-2′/H-6′, and H-3′/H-5′, and of *δ_H_* 7.23 (1H, *t*, *J* = 7.6 Hz, H-4′) corresponding to *δ_C_* 130.7 (C-2′/C-6′), *δ_C_* 129.7 (C-3′/C-5′), and *δ_C_* 127.8 (C-4′) of the monosubstituted phenyl group, respectively. In addition, the HMBC correlations of H-2′/H-6′ to *δ_C_* 43.5 (C-10) and from H_2_-10 to *δ_C_* 137.9 (C-1′), C-2′/C-6′, and C-3 resulted in the attachment of the monosubstituted phenyl group to the perhydroisoindolone residue forming the 10-phenylperhydroisoindolone moiety. The remaining portion of **1** was identified as a 13-membered carbocycle ring, which was elucidated by a combination of homonuclear COSY cross peaks establishing a 13-proton spin system resonating from H-7 to H-22 and based on the key HMBC correlations from *δ_H_* 6.01 (1H, *ddd*, *J* = 15.2, 9.6, 1.9 Hz, H-13) to *δ_C_* 40.5 (C-15), from *δ_H_* 1.09 (1H, *m*, H-17a) and *δ_H_* 1.26 (1H, *m*, H-17b) to *δ_C_* 32.6 (C-19), from *δ_H_* 1.39 (1H, *m*, H-18b) to *δ_C_* 76.4 (C-20), and from *δ_H_* 6.48 (1H, *dd*, *J* = 15.4, 8.5 Hz, H-21) and *δ_H_* 6.86 (1H, *d*, *J* = 15.4 Hz, H-22) to *δ_C_* 198.6 (C-23). Finally, HMBC correlations from H-13 to *δ_C_* 51.9 (C-8) as well as the long-range coupling from H-4 and H-22 to C-23 suggested the connectivity of the 13-membered carbocyclic ring residue to the 10-phenylperhydroisoindolone moiety.

Based on the COSY and HMBC correlations discussed above, the proton and carbon signals originating from the cytochalasin core structure and the macrocyclic ring fragment were found to have a close resemblance to deoxaphomin [4,29]. The main difference is an additional carboxylic carbon resonating at *δ_C_* 171.6 (C-25) and a methyl carbon at *δ_C_* 21.1 (C-26) and *δ_H_* 2.04 (3H, *s*, H-26). Furthermore, the NMR signals of the acetyl group were evidenced by HMBC correlations from *δ_H_* 5.20 (1H, *m*, H-20) to C-25 and from H-26 to C-25, indicating the attachment of the acetyl group at C-20.

The relative configuration of **1** was deduced through a combination of proton coupling constants and nuclear Overhauser effect spectroscopy (NOESY) data (Appendix A). In the ^1^H NMR spectrum, the large coupling constant (*J* = 10.0 Hz) between H-7 and H-8 suggested coaxial positioning of this proton pair. The *trans*-geometric configurations of Δ^13(14)^ and Δ^21(22)^ were established based on the observed large coupling constants (H-13, H-14, *J* =15.1 Hz; H-21, H-22, *J* = 15.3 Hz). Additional NOESY interactions of H-3 with H_3_-11 and H-7 revealed similar spatial orientation for C-3 benzyl and C-7 hydroxy moieties, whereas correlations of H-4 with H-5 and H-8 were *β*-oriented. Based on biosynthetic considerations and previous studies [30,31] naturally occurring cytochalasins possess similar *α*-orientation of the methyl groups attached to C-16, whereas the hydroxyl group at C-20 is *β*-oriented. Thus, the relative configuration of 16-Me and 20-OH in **1** may follow that of known and related cytochalasin derivatives.

Triseptatin (**1**) was also subjected to ECD– TDDFT calculations to confirm its absolute configuration. MMFF94 conformational analysis and B3LYP/6-31G(d) geometry reoptimization of **1** afforded two energetically low-lying conformers with a preassigned (3*S*,4*S*,6*S*,6a*R*,7*E*,10*R*,14*R*,15*E*,17a*R*) configuration (Figure 3). All conformers were confirmed stable as per harmonic vibrational frequency calculations. The differences in spatial structural arrangement of compound **1** conformers were a result of the C3–C10 bond rotation, with the ωC-1′,C-10,C-3,C-4 torsion angles for conformers 1A (78.79%) and 1B (21.21%) being 69.38° and −177.72°, respectively. The theoretically obtained Boltzmann-averaged ECD spectrum of **1** at wB97XD/DGDZVP (PCM/MeCN) after wavelength correction showed a good correlation with the experimental data (Figure 4), confirming that the absolute configuration of triseptatin is (3*S*,5*S*,7*S*,8a*R*, 9a*R*, 13*E*,16*R*,20*R*,21*E*)–**1**.

Triseptatin (**1**) and deoxaphomin B (**2**) were biologically tested for their cytotoxicity in seven mammalian cell lines, including the mouse fibroblast line (L929), HeLa carcinoma line (KB3.1), human breast adenocarcinoma cell line (MCF-7), human lung carcinoma line (A549), human prostate cancer (PC-3), ovarian carcinoma cell line (SKOV-3), and squamous cell carcinoma (A431), using MTT assay with epothilone B as a positive control. Cytochalasans **1** and **2** exhibited cytotoxic activities against all cell lines with half-maximal inhibitory concentrations of 1.80 to 11.28 μM and 1.55 to 6.91 μM, respectively (Table 2). While comparable IC_50_ values were noted against most cancer cell lines, compound **2** exhibited five times potency versus the prostate cancer cells. However, the cytotoxic activity of the isolated cytochalasans were relatively weak in comparison to the half-maximal inhibitory concentration values obtained from epothilone B. Compounds **1**–**3** were also assessed for antiproliferative activity against human umbilical vein epithelial cell (HUVEC) and myelogenous leukemia cell (K-562) and cytotoxicity against HeLa cervical cancer cell line through CellTiter Blue assay (Table 2). All compounds exhibited antiproliferative effects and cytotoxic activities. Compounds **2** and **3** were cytotoxic against HeLa cells with IC_50_ values of 4.96 and 7.30 µM, respectively. Cytochalasins **1**–**3** showed moderate antileukemic activity, while strong antiproliferative effects were observed against HUVEC cells.

Compounds **1**, **2**, **3**, and cytochalasin D (**6**) were evaluated in 1 h endpoint assays with selected concentrations estimated based on previously determined half-maximal inhibitory concentrations, referred to as low dose (LD, 1 × IC_50_) and high dose (HD, 5 × IC_50_) for their impact on the F-actin network of U2-OS cells. F-actin-containing structures were monitored by staining with fluorescently coupled phalloidin (seen in red, Figure 5 and Figure 6). LD of cytochalasin B and D (Figure 5c,d) led to F-actin networks devoid of lamellipodial structures with F-actin-rich protrusions at the cell periphery clearly visible in the DMSO vehicle control (Figure 5i) and F-actin-rich knot-like structures in the main cell body. Cytochalasin D (Figure 5h) in HD further increased the number of these aggregates, coinciding with a reduction in other visible F-actin-containing structures in the main cell body, while HD of cytochalasin B (Figure 5g)-treated cells showed a severe impediment but no total collapse of the F-actin network. The impact of **1** (Figure 5a,e) in this set of experiment most closely resembled the effect induced by cytochalasin B intoxication and was characterized by a lack of lamellipodial structures and the presence of F-actin-rich aggregates in the main cell body. This observation was much more pronounced in the high-dose experiment (Figure 5e), coinciding with a lower number of visible stress fibers. Deoxaphomin B (**2**) applied in LD (Figure 5b), however, led to strongly contrasted cable-like remnants in comparison to the dim fluorescence signal of phalloidin detected in the main cell body. A high dose (Figure 5f) led to a well-described endpoint of cytochalasin intoxication, with F-actin-containing aggregates dominating the otherwise collapsed F-actin network with no visible stress fibers. The impact of the tested compounds on the F-actin organization was fully reversible in all cases (compare Figure 5e and Figure 5j, Figure 5f and Figure 5k, Figure 5g and Figure 5l, and Figure 5h and Figure 5m with DMSO vehicle controls Figure 5i and Figure 5n). Concurrently, the concentration-dependent effect of the cytochalasans on the F-actin cytoskeletal organization was examined incrementally (note that compound **1** was only assessed in a high-dose/low-dose assay due to scarcity of the compound). First, changes to the F-actin network presented themselves by a reduction in lamellipodia-like structures and formation of previously described phalloidin-stainable F-actin-containing aggregates (**2**: Figure 6d, 1 µM; **3**: Figure 6l, 1 µM; and **6**: Figure 6r, 0.1 µM). The effects did not grow more severe once a concentration was reached, which led to a full collapse of F-actin-containing macrostructures, with the F-actin aggregates remaining in the cell (**2**: Figure 6g, 30 µM; **3**: Figure 6n, 10 µM; and **6**: Figure 6l, 1 µM). Remarkably, the progression of F-actin network dispersal was different among cells treated with deoxaphomin B and cytochalasin B. Here, the periphery of the cells remained largely intact with strongly contrasted stress-fiber-like structures appearing at higher concentrations as had been observed before, which did not appear at any tested concentration after cytochalasin B and D treatment (compare Figure 6d–f with Figure 6m–o and Figure 6k–w).

The observations fit well with the well-described phenotype of cytochalasin-induced F-actin disorganization [32,33]. Notably, deoxaphomin B seemed to have a stronger effect on stress fiber organization even in low doses compared to the other cytochalasans tested here. The reason for this behavior has to be scrutinized in the future once more material of the compound becomes available.

The deoxaphomin B reported here differed from deoxaphomin by the presence of a double bond inside the six-ring between C-5 and C-6 instead of deoxaphomin with a methylidene group of C-12, while triseptatin featured an additional O-acetyl group at the C-20 position. Interestingly, desoxaphomin was reported to exhibit irreversible effects on the F-actin organization of U2-OS cells, while the triseptatin tested here and the closely related deoxaphomin B were both shown to exhibit fully reversible effects even when applied in high concentrations [6]. Notably, Lambert et al. [7] reported that the newly described pseudofuscochalasin A exhibited irreversible effects, while cells treated with cytochalasin C, which differ from the former by an additional acetyl group at the ketone at C-23, were able to fully recover after one hour. The results presented here indicate that the α–β unsaturated bond next to the ketone should not be considered the solely decisive factor steering an unrecoverable impact on F-actin organization. To further understand the phenomenon of reversible and irreversible changes towards the F-actin network induced by cytochalasins, two emerging hypotheses have to be examined in the future: (1) the configuration of the six-ring impacting reversibility and (2) the stereochemistry of the hydroxy group at C-20 also exhibiting influences on reversibility. Recent findings comparing cytochalasin E derivatives indicated that a change from a methylidene group to two methyl groups influencing the stereochemistry at the six-ring found in cytochalasin K had a profound impact on its cytotoxicity, being 80 times less toxic than the former. Moreover, the effect was found to be partially reversible without observation of a full recovery after one hour (C. L. et al., unpublished data). Deoxaphomin B features the previously mentioned methylidene group but exhibits acetylation at the hydroxyl group with unknown consequences for the bioactivity in question. In the light of the current findings, hypothesis 1 seems more tempting, but further research focusing on finding a producer of a triseptatin featuring a C-20-deacetyl deoxaphomin B backbone is necessary to formally exclude hypothesis 2.

## 4. Conclusions

This study described the biologically active chemical constituents of the rice fermentation culture of *Sparticola triseptata*, where one new cytochalasin derivative, triseptatin (**1**), along with deoxaphomin B (**2**), cytochalasin B (**3**), *cis*-4-hydroxy-6-deoxyscytalone (**4**), and 6-hydroxymellein (**5**), were isolated and identified. Compounds **1** and **2** displayed cytotoxic activities against seven mammalian cell lines (L929, KB3.1, MCF-7, A549, PC-3, SKOV-3, and A431) and antiproliferative effects against myeloid leukemia K-562 cancer cell line. Further investigation on the mechanism of their cytotoxicity showed the reversible effects of cytochalasans **1**–**3** on the disruption of the actin cytoskeleton in vivo. Our findings indicate that the presence of α–β unsaturated bond adjacent to the ketone group at C-23 does not directly correlate with the reversibility of this effect. Rather, two previously unrecognized structural features of the cytochalasin framework could have significant impact on the reversible actin disruption, namely the configuration of the cyclohexene ring and the chirality of the hydroxy group attached at C-20. In general, we have established the potential of cytochalasin derivatives as a possible drug inspiration for anticancer agents.

## Figures and Tables

**Figure 1 jof-08-00560-f001:**
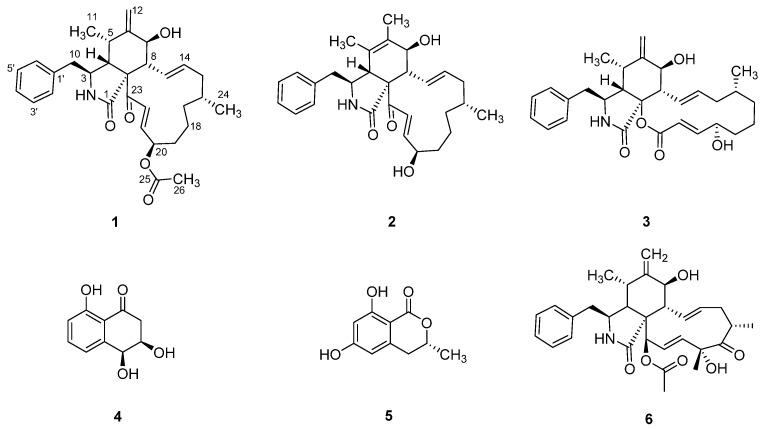
Secondary metabolites **1**–**5** from *Sparticola triseptata* and cytochalasin D (**6**).

**Figure 2 jof-08-00560-f002:**
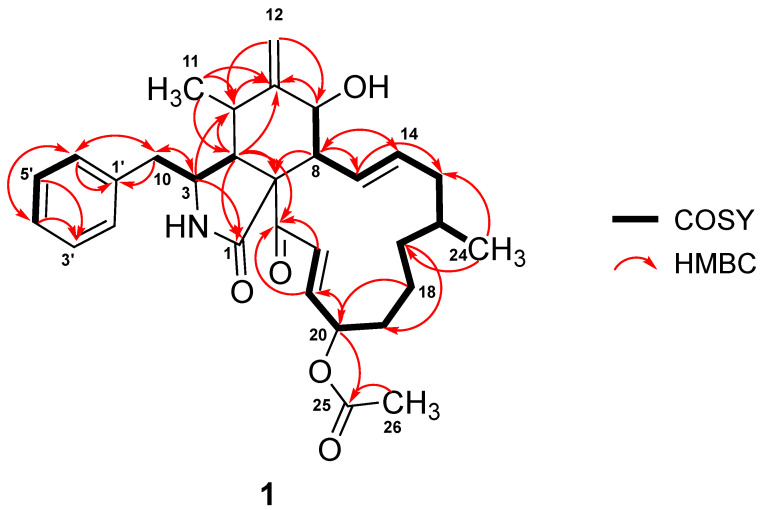
COSY and HMBC correlations in **1**.

**Figure 3 jof-08-00560-f003:**
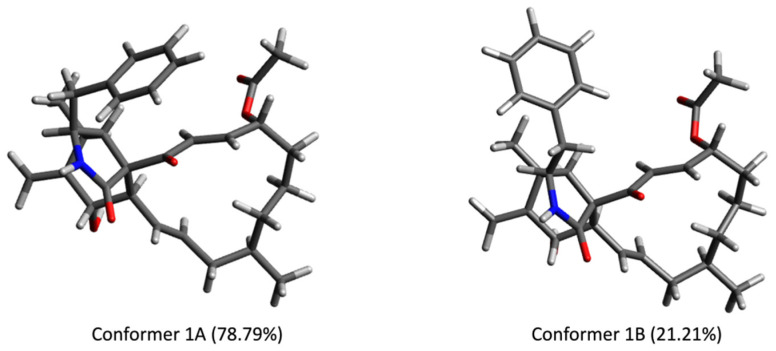
Low-energy conformers (>1%) of (3*S*,5*S*,7*S*,8a*R*, 9a*R*,13*E*,16*R*, 20*R*,21*E*)-**1** optimized at B3LYP/6-31G(d) (PCM/MeOH).

**Figure 4 jof-08-00560-f004:**
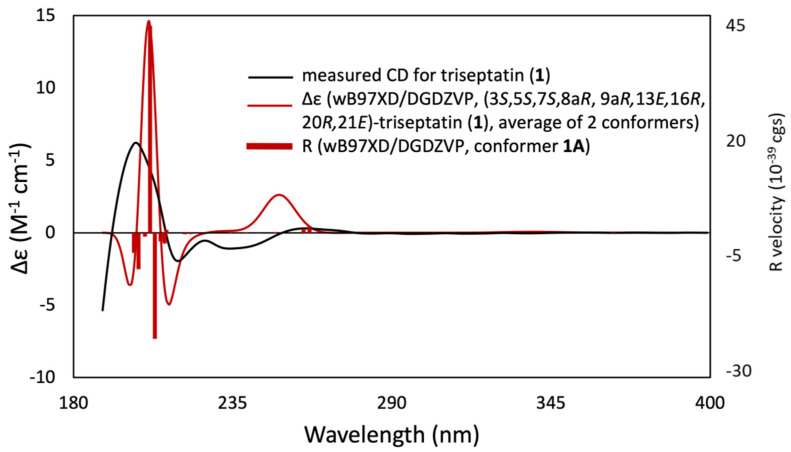
Experimental ECD spectrum of triseptatin (**1**, black solid curve) compared with wB97XD/DGDZVP-calculated ECD spectra (red solid curve) for the B3LYP/6-31G(d)-optimized conformers of (3*S*,5*S*,7*S*,8a*R*, 9a*R*, 13*E*,16*R*,20*R*,21*E*)–**1**.

**Figure 5 jof-08-00560-f005:**
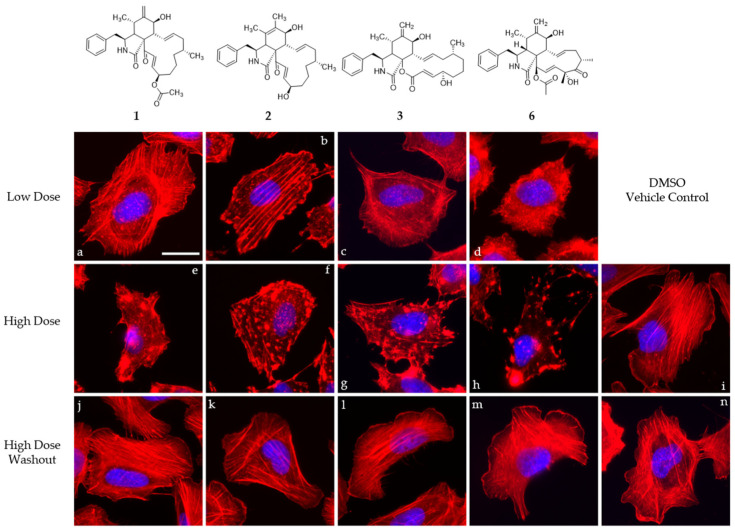
Overlay images of immobilized U2-OS cells treated with different concentrations of compounds **1**–**3** and **6** (**1**: (**a**,**e**,**j**), **2**: (**b**,**f**,**k**), **3**: (**c**,**g**,**l**), and **6**: (**d**,**h**,**m**)) normalized against their cytotoxicity against L929 cells for one hour (IC_50_; low dose, 1 × IC_50_ (**a**–**d**); high dose, 5 × IC_50_ (**e**–**h**)) and the corresponding high-dose washout experiment after one hour recovery time (**j**–**m**). Volume of the DMSO vehicle control was adjusted to the highest volume of DMSO used in the corresponding experiment (**i**,**n**). Cells were stained for their F-actin cytoskeleton using fluorescently coupled phalloidin (pseudocoloured in red) and nuclear DNA using DAPI (pseudocoloured in blue). Representative scale bar in image (**a**) correspond to 25 µm.

**Figure 6 jof-08-00560-f006:**
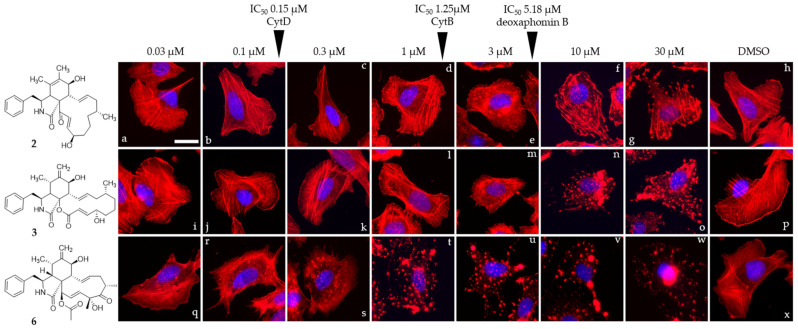
Overlay images of immobilized U2-OS cells treated with a gradually increasing concentration (0.03–30 µM) of compounds **2**, **3**, and **6** (**2**: (**a**–**g**), **3**: (**i**–**o**), and **3**: (**q**–**w**)) for one hour. IC_50_ values determined against L929 cells are denoted with arrows. Volume of the DMSO vehicle control was adjusted to the highest volume of DMSO used in the corresponding experiment (**h**,**p**,**x**). Cells were stained for their F-actin cytoskeleton using fluorescently coupled phalloidin (pseudocoloured in red) and nuclear DNA using DAPI (pseudocoloured in blue). Representative scale bar shown in image (**a**) represent 25 µm.

**Table 1 jof-08-00560-t001:** NMR spectroscopic data of **1** in MeOH-*d*_4_.

Position	1	
	*δ**_H_* (mult., *J* in Hz)	*δ_C_*, Type
1	-	176.4, C
3	3.38 (m)	54.0, CH
4	2.91 (m)	46.3, CH
5	2.72 (m)	33.2, CH
6	-	151.0, C
7	3.90 (d, 9.9)	72.8, CH
8	2.50 (t, 9.9)	51.9, CH
9	-	63.9, C
10a	2.46 (dd, 13.0, 7.3)	43.5, CH_2_
10b	2.64 (dd, 13.0, 6.4)	
11	0.79 (d, 6.8)	13.3, CH_3_
12a	5.03 (s)	114.1, CH
12b	5.17 (s)	
13	6.01 (ddd, 15.2, 9.6, 1.9)	128.3, CH
14	5.22 (m)	137.3, CH
15a	2.06 (m)	40.5, CH_2_
15b	1.79 (dd, 11.2, 2.8)	
16	1.51 (m)	34.0, CH
17a	1.09 (m)	34.9, CH_2_
17b	1.26 (m)	
18a	1.15 (m)	19.7, CH_2_
18b	1.39 (m)	
19a	1.52 (m)	32.6, CH_2_
19b	1.92 (m)	
20	5.20 (m)	76.4, CH
21	6.48 (dd, 15.4, 8.5)	144.0, CH
22	6.86 (d, 15.4)	128.0, CH
23	-	198.6, C
24	0.89 (d, 6.9)	20.8, CH_3_
25	-	171.6, C
26	2.04 (s)	21.1, CH_3_
1′	-	137.9, C
2′/6′	7.08 (d, 7.6)	130.7/131.0, CH
3′/5′	7.32 (t, 7.6)	129.7/129.6, CH
4′	7.23 (t, 7.6)	127.8, CH

Recorded at 600 MHz; carbon multiplicities were deduced from HSQC–DEPT-135 spectra.

**Table 2 jof-08-00560-t002:** Antiproliferative effect and cytotoxicity of **1**–**2** against mammalian cell ines.

Cell Line	Compound			Positive Control	
	1	2	3	Epothilone B	Imatinib
Cytotoxicity ^a^ IC_50_ (μM)					
Mouse fibroblast L929	4.16	5.18	N.D.	1.4 × 10^−^^3^	N.D.
HeLa cells KB3.1	1.80	1.83	N.D.	8.9 × 10^−^^5^	N.D.
Human breast adenocarcinoma MCF-7	1.86	1.79	N.D.	2.4 × 10^−^^4^	N.D.
Human lung carcinoma A549	7.32	6.91	N.D.	6.9 × 10^−^^5^	N.D.
Human prostate cancer PC-3	11.28	2.81	N.D.	1.6 × 10^−^^3^	N.D.
Ovarian carcinoma SKOV-3	1.84	1.55	N.D.	2.8 × 10^−^^4^	N.D.
Squamous cell carcinoma A431	2.17	1.60	N.D.	7.9 × 10^−^^5^	N.D.
Antiproliferative Effect ^b^ GI_50_ (µM)					
HUVEC	4.55	1.08	8.31	N.D.	18.5
Myelogenous leukemia K-562	8.31	3.67	3.34	N.D.	0.17

^a^ MTT assay, ^b^ CellTiter blue assay. The starting concentration for the cytotoxicity assay was 300 μg/mL, and substances were dissolved in MeOH (1 mg/mL). MeOH was used as the negative control and showed no activity against the tested mammalian cell lines. Results were expressed as IC_50_: half-maximal inhibitory concentration, CC_50_: half-maximal cytotoxicity concentration, GI_50_: half-maximal cell proliferation (μM), N.D: not determined.

## Data Availability

Not applicable.

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
