# Peer review of "Antiproliferative and Cytotoxic Cytochalasins from *Sparticola triseptata* Inhibit Actin Polymerization and Aggregation"

_jof, 2022, doi:10.3390/jof8060560_

Round 1

Reviewer 1 Report

In this manuscript one more cytochalasin (triseptatin) was described. Its cytotoxicity and inhibitory activity on actin polymerization and aggregation was compared to some other well-known cytochalasins. There are some points to address before publication:

1) check spelling, e.g.

line 59 (formerly also known as Phomopsis, and Xylaria

line 81 structure elucidation of one cytotoxic cytochalasin derivatives

line 310 were biologically tested for their cytotoxicity

2) line 70 add references after "Previously we reported the actin depolymerization activity of 25 cytochalasans ... "

3) compare results of cytotoxicity bioassays with positive control treatments

4) what was a positive control treatment in the actin polymerization and aggregation assay?

5) It would be good to stress the novelty and importance of the manuscript as well as the research prospects in the conclusion

Author Response

Reviewer 1

Comments

Revisions

1) check spelling, e.g.

line 59 (formerly also known as Phomopsis, and Xylaria

line 81 structure elucidation of one cytotoxic cytochalasin derivatives

line 310 were biologically tested for their cytotoxicity

line 59. …(formerly known as Phomopsis, and Xylaria)

line 80 structure elucidation of one cytotoxic cytochalasin derivative

line 310 retain

2) line 70 add references after "Previously we reported the actin depolymerization activity of 25 cytochalasans ... "

Line 70

"Previously we reported the actin depolymerization activity of 25 cytochalasans ... " [5]

3) compare results of cytotoxicity bioassays with positive control treatments

Line 321 – 323:

However, the cytotoxic activity of the isolated cytochalasans were relatively weak in comparison to the half-maximal inhibitory concentration values obtained from Epothilone B.

4) what was a positive control treatment in the actin polymerization and aggregation assay?

For the actin assays, DMSO was used as the vehicle control serving as a negative control, while the comparison with the well studied cytochalasin B and D served as the positive controls. Both are studied and used as tool compounds in cell biology for decades to study F-actin dependent effects due to their ability to inhibit F-actin polymerization due to their well characterized ability to bind to the barbed end of growing actin filaments. As a reference, the reviewer can study the papers from Cooper 1987 (https://pubmed.ncbi.nlm.nih.gov/3312229/) or Yahara et al. 1982 (https://pubmed.ncbi.nlm.nih.gov/7199054/) and a more recent paper (Kretz et al. 2019, https://www.ncbi.nlm.nih.gov/pmc/articles/PMC6406453/). We feel that our description of using cytochalasins as well described examples (p. 10 l. 359) for the cytochalasin induced phenotype clearly state our intention to discuss cytochalasin B and D as positive controls.

5) It would be good to stress the novelty and importance of the manuscript as well as the research prospects in the conclusion

Line 416 – 430:

4. Conclusion

This study described the biologically active chemical constituents of the rice fermentation culture of Sparticola triseptata, where one new cytochalasin derivative, triseptatin (1), along with deoxaphomin B (2), cytochalasin B (3), cis-4-hydroxy-6-deoxyscytalone (4), and 6-hydroxymellein (5) were isolated and identified. Compounds 1 and 2 displayed cytotoxic activities against seven mammalian cell lines (L929, KB3.1, MCF-7, A549, PC-3, SKOV-3, A431) and antiproliferative effects against myeloid leukemia K-562 cancer cell line. Further investigation on the mechanism of their cytotoxicity showed the reversible effects of cytochalasans 1 – 3 on the disruption of the actin cytoskeleton in vitro. Our findings indicate that the presence of α-β unsaturated bond adjacent to the ketone group at C-23 does not directly correlate with the reversibility of this effect. Rather, two structural features of the cytochalasin framework could have significant impact on the reversible actin disruption which are the configuration of the cyclohexene ring, and the chirality of the hydroxy group attached at C-20. In general, we have established the potential of cytochalasin derivatives as a possible drug inspiration for anti-cancer agents.

Reviewer 2 Report

In this manuscript a new cytochalasan triseptatin (1), along with previously described cytochalasans deoxaphomin B (2) and cytochalasin B (3), were isolated from the rice culture of S. triseptata. The structure of 1 was elucidated. The absolute configurations were established through analysis of NOESY and ECD-TDDFT. Compounds 1 and 2 showed cytotoxic activities against seven mammalian cell lines and antiproliferative effects against myeloid leukemia K-562 cancer cell line. Both 1 and 2 could be shown to possess F-actin network inhibiting properties. The manuscript was well prepared. The results are interesting. I recommend to accept it for publication. Some type and grammar errors need to be revised, such as: 

page 2 line 66: has→have;

page 2 line 81: derivatives→derivative;

page 6 line 276: indicated→indicating;

page 7 line 298: add "a" before "good correlation";

page 9 line 314: positive control→a positive control;

page 10 line 348: lead to→led to;

page 10 line 356: were→was;

page 10 line 358: itself→themselves;

page 12 line 393: keton→ketone;

page 12 line 406: delete "an".

Author Response

Reviewer 2

Comments

Revisions

page 2 line 66: has→have;

biological effects of cytochalasans have been related to their interference…

page 2 line 81: derivatives→derivative;

Line 80

structure elucidation of one cytotoxic cytochalasin derivative

page 6 line 276: indicated→indicating;

Line 279

indicating the attachment of the acetyl group at C-20.

page 7 line 298: add "a" before "good correlation";

Line 301

showed a good correlation with the experimental data

page 9 line 314: positive control→a positive control;

Line 317

with epothilone B as a positive control

page 10 line 348: lead to→led to;

Line 344

however, led to strongly contrasted cable

page 10 line 356: were→was;

Line 350 – 352 

Concurrently, the concentration dependent effect of the cytochalasans impact on the F-actin cytoskeletal organization was examined incrementally

page 10 line 358: itself→themselves;

Line 353

presented themselves by a reduction

page 12 line 393: keton→ketone;

Line 398

at the ketone at C-23

page 12 line 406: delete "an".

Line 411

exhibits acetylation

Reviewer 3 Report

In the manuscript (jof-1723539) by Garcia and coworkers, a novel cytochalasin analog triseptatin along with four known compounds were isolated from the rice culture of S. triseptata. The chemical structure of triseptatin was fully characterized by spectroscopic methods as well as ECD calculation using TDDFT. Two compounds showed cytotoxicity and antiproliferation effects in the cell-based screening models. Additionally, these two compounds were found to inhibit F-actin network and the plausible reason for their reversible property were discussed. This work is interesting and will expand the understanding of the material basis of cytochalasins on actin. Meanwhile, several points need be addressed before publication.

  • Please check the correctness of this description, P2, “on a Bruker Ascend 700 MHz spectrometer equipped with a 5-mm TXI cryoprobe (1H 600 MHz, 13C 600 MHz)”.
  • The chirality on C4 need be labelled clearly.
  • HPLC-ECD data (Φ) should be ‘Δε’.
  • The data of positive control need be represented in the unit of micromolar.
  • In Figure 4, the baselines of theoretical and experimental ECD spectra should be overlapped.
  • Figures in SI need be much clearer.

Author Response

Reviewer 3

Comments

Revisions

Please check the correctness of this description, P2, “on a Bruker Ascend 700 MHz spectrometer equipped with a 5-mm TXI cryoprobe (1H 600 MHz, 13C 600 MHz)”.

on a Bruker Ascend 600 MHz spectrometer equipped with a 5-mm TXI cryoprobe (1H 600 MHz, 13C 150 MHz)

The chirality on C4 need be labelled clearly.

Page 5. Figure 1 was revised accordingly.

HPLC-ECD data (Φ) should be ‘Δε’.

Line 148:

HPLC-ECD data in acetonitrile as lmax (F) 205 (24.4), 216 (–8.2), 239 (–3.5).

is changed to:

HPLC-ECD data in acetonitrile as lmax (Δε) 205 (0.0074), 216 (–0.0025), 239 (–0.011).

The data of positive control need be represented in the unit of micromolar.

Epothilone B and Imatinib are now expressed in mM.

In Figure 4, the baselines of theoretical and experimental ECD spectra should be overlapped.

Figure 4 was revised to correct the baselines.

Figures in SI need be much clearer.

Supplementary Material was revised.

Reviewer 4 Report

The present manuscript “Antiproliferative and cytotoxic cytochalasin from Sparticola  triseptata inhibit actin polymerization and aggregation” provides valuable information on newly identified compound and its disruptive activity on mammalian cell actin. The chemical part and bioactivity tests of the manuscript are detailed, and the article is well constructed and written. Comments are listed as follows.

  1. Line 87  “Bruker Ascend 700 MHz” should be “ Broker Ascend 600 MHz”. 
  2. Line 88 “13C 600M Hz” should be “ 13C 150 MHz”.
  3. Line 107 - line 110, what is the temperature for the two fermentation processes?  Please detail it. 
  4. Line 127 - line 128, line 137 - line 138, what is solvent B? What is its composition? Detail it please.
  5. Line 146, the error between calculated value 506.2828 and founded value 506.2901 of the [M + H ] is 0.7%, more than the standard value 0.5%. Please explain. Please it's better to re-test the HRESIMS.
  6. Recommend deletion the NOE correlations from Figure 2.
  7. Table 1, positon 26, “21.1” should be “21.1 CH3”. 

Author Response

Reviewer 4

Comments

Revisions

Line 87: “Bruker Ascend 700 MHz” should be “ Broker Ascend 600 MHz”. 

Bruker Ascend 600 MHz spectrometer equipped with a 5-mm TXI cryoprobe (1H 600 MHz, 13C 150 MHz)

Line 88 “13C 600M Hz” should be “ 13C 150 MHz”.

Corrected to 150 MHz

Line 107 - line 110, what is the temperature for the two fermentation processes?  Please detail it. 

Temperature was already stated po

Line 110 – 112:

Five agar blocks of a well-grown fungal culture were inoculated in the culture flasks and incubated under static condition in a dark room at 25–30 °C for 12 weeks until the fungal hyphae proliferated and the rice medium turned black in colour. 

Line 127 - line 128, line 137 - line 138, what is solvent B? What is its composition? Detail it please.

Line 129 – 131:

The third subfraction, Fraction 3.2.3 (259 mg) was further purified using semi-preparative RP-HPLC. The mobile phase is composed of ultrapure water (Milli-Q, Millipore, Schwalbach, Germany; solvent A) and acetonitrile (RCI Labscan Ltd., HPLC grade; solvent B).

Line 146, the error between calculated value 506.2828 and founded value 506.2901 of the [M + H ] is 0.7%, more than the standard value 0.5%. Please explain. Please it's better to re-test the HRESIMS.

Line 150: Typographical error on the calculated value of C31H40NO5:

HR-ESI-MS m/z: [M + H]+ calcd for C31H40NO5, 506.2906; found, 506.2901.

Recommend deletion the NOE correlations from Figure 2.

Deleted NOE correlations

Table 1, position 26, “21.1” should be “21.1 CH3”. 

Table 1, position 26, “21.1, CH3

Reviewer 5 Report

Although a new cytochalasan analog triseptatin (1) along five known ones were isolated and identified from the rice culture of S. triseptata, the novelty of the compounds is poor.

1) the new compound 1 seems to be an artifact with a CH3COO-group at C20. IF this is not an artifact, please give the experimental confirmation.

2)ECD calculation in figure 4 should be improved. ECD of calculated two conformers and experimental one should be shown in one figure.

Author Response

Reviewer 5

Comments

Revisions

1) The new compound 1 seems to be an artifact with a CH3COO-group at C20. If this is not an artifact, please give the experimental confirmation.

Acetylated derivatives are common in many reported cytochalasin derivatives. Fermentation was carried out using standard protocols practiced in our laboratories with no known acetylating inducing conditions. For purification, we used   pure, unacidified solvents.

2) ECD calculation in Figure 4 should be improved. ECD of calculated two conformers and experimental one should be shown in one figure.

Figure 4 was revised to correct the baselines. The calculated ECD spectra is presented as a Boltzmann-weighted average of the two conformers. The experimental ECD is also shown in the same figure.

Round 2

Reviewer 1 Report

The manuscript seems ready for publication

Reviewer 5 Report

I recommend accepting it in its present form.